# The Portiloop: A deep learning-based open science tool for closed-loop brain stimulation

**Nicolas Valenchon**[1], **Yann Bouteiller**[1]*, **Hugo R. Jourde**[2], **Xavier L'Heureux**[1], **Milo Sobral**[1], **Emily B. J. Coffey**[2], **Giovanni Beltrame**[1]

**1** MISTLab—Polytechnique, University of Montreal, Montreal, Quebec, Canada, **2** CL:ASP, Concordia University, Montreal, Quebec, Canada

* yann.bouteiller@polymtl.ca

**Data Availability Statement:** The data underlying the results presented in this study are available from our GitHub along with our code and hardware plans: https://github.com/Portiloop.

## Abstract

Closed-loop brain stimulation refers to capturing neurophysiological measures such as electroencephalography (EEG), quickly identifying neural events of interest, and producing auditory, magnetic or electrical stimulation so as to interact with brain processes precisely. It is a promising new method for fundamental neuroscience and perhaps for clinical applications such as restoring degraded memory function; however, existing tools are expensive, cumbersome, and offer limited experimental flexibility. In this article, we propose the Portiloop, a deep learning-based, portable and low-cost closed-loop stimulation system able to target specific brain oscillations. We first document open-hardware implementations that can be constructed from commercially available components. We also provide a fast, lightweight neural network model and an exploration algorithm that automatically optimizes the model hyperparameters to the desired brain oscillation. Finally, we validate the technology on a challenging test case of real-time sleep spindle detection, with results comparable to off-line expert performance on the Massive Online Data Annotation spindle dataset (MODA; group consensus). Software and plans are available to the community as an open science initiative to encourage further development and advance closed-loop neuroscience research [https://github.com/Portiloop].

## 1 Introduction

Electrical activity within the brain forms the basis of perception, thought and behaviour. This activity tends to be oscillatory in nature, as reciprocal connections within and between brain regions form functional circuits for processing and communicating information. Changes in electrical fields caused by synchronously firing populations of neurons can be measured on the scalp using a technique known as electroencephalography (EEG). Correlational studies have been performed for nearly a century that attempt to link specific patterns and frequency bands in EEG to cognitive functions or brain states. These approaches are informative for many types of research questions and have increased our understanding of brain processes, but they are unable to establish causal relationships. The ability to interact with brain oscillations in a precisely-timed fashion to enhance or inhibit endogenous processes—using sensory [1–4],

**Funding:** This work was supported by the AUDACE grant, awarded to G.B and E.C. by the Quebec Research Fund (FRQSC 279561 - https://frq.gouv.qc.ca) and approved by Polytechnique Montreal (CER-2021-39-R - https://www.polymtl.ca). The funders had no role in study design, data collection and analysis, decision to publish, or preparation of the manuscript.

**Competing interests:** The authors have declared that no competing interests exist.

electrical [5] or magnetic [6] stimulation—allows for their functional roles to be determined [7], and potentially for restoration of processes deteriorated by aging or pathology [8]. These so-called *closed-loop* stimulation approaches thus hold great promise for neuroscience.

One of the closed-loop research areas that has progressed the fastest using non-invasive neurophysiological recordings (i.e., EEG) and brain stimulation techniques is studying memory consolidation processes in sleep [1, 9–11]. A first target has been slow oscillations (SOs: 0.5—1.5 Hz), which are high amplitude waves that appear in non-rapid eye movement (NREM) sleep and are known to be involved in memory consolidation (i.e., the process by which recent learned experiences are transformed into long-term memory) [1]. Using auditory stimulation to SO up-states, when neural tissue is partly depolarized and more excitable, Ngo *et al.* enhanced the amplitude of SOs and reported an overnight improvement in memory performance, a result that has now been replicated multiple times (see [12–14] for reviews). Closed-loop stimulation has also been used in the context of preventing drowsiness [2], enhancing attention and engagement [4], and reducing central nervous system damage after strokes [3]. There is great potential for these closed-loop stimulation techniques in fundamental neuroscience, and potentially, for clinical applications [7, 15]. However, progress is hampered by the limited portability and flexibility of available systems, as well as by their expense and by the complexity of their use.

The goal of our interdisciplinary collaboration between neuroscientists, data scientists and computer engineers is to design, explore, and document the properties of a new, complete closed-loop stimulation system (i.e., hardware and software), which we call the *Portiloop*. The Portiloop is a deep learning-based, portable, battery-efficient and low-cost device that will enable the neuroscience community to collect and process EEG data in real-time, detect patterns of interest for fundamental research questions, and respond at low latency with precisely-timed stimulation. We aim to accelerate fundamental research on closed-loop stimulation in neuroscience by designing a highly functional device and offering the code and plans freely to developers and scientists in the research community.

The scope of this work encompasses both neuroscience and engineering aspects, which may be of interest to audiences for different purposes. First, we describe some general background concerning the use of closed-loop stimulation in neuroscience and its potential, describe limitations in existing tools, and introduce sleep spindles, a fast neural event that is observable in EEG, as a challenging test case. Next, we discuss the real-time and portability design constraints and the (hardware) architecture of our Portiloop implementation, which is sufficiently powerful to allow us to run a neural network-based EEG detection algorithm. The hardware is not commercially available in assembled state, but it or a similar device may be constructed by readers with appropriate technical training (plans and additional information are freely available [https://github.com/Portiloop]). Third, we describe a lightweight neural network architecture that can run on inexpensive, modest hardware systems such as that which we have proposed, and which can detect and react to physiological signals in real time. Most importantly, we detail our design methodology and optimization algorithm, so that the architecture can be adapted to other neural events (e.g., theta or beta-band oscillations) or types of signal (e.g., functional near-infrared spectroscopy). This latter section and associated S1 File will be of most interest to readers with a data science background who may wish to implement, use or modify the detection algorithm (all code is available). We then present data from our case study EEG event, showing that the Portiloop implementation can effectively detect sleep spindles in real time, and we describe the performance with respect to detection threshold and time delay. The latter sections may be most interesting for research users to understand the performance of the system and select appropriate parameters for its use

detecting and stimulating brain oscillations. Finally, we discuss next steps and future prospects for this technology.

The Portiloop is the first open-science device that is capable of closed-loop brain stimulation. Its most noteworthy contributions include:

- Two open-hardware implementations that can be constructed from commercially available components (one using the Xilinx Pynq FPGA together with the HackEEG board and one using a custom board and a Google Coral System-on-Module neural accelerator)

- A fast implementation of a recurrent neural network model that can be run on inexpensive hardware to detect events in physiological signals in real time

- A design-space exploration algorithm that automatically optimizes the model hyperparameters to the neural event to be detected

- A real-time spindle detector with accuracy comparable to offline analysis by experts

We hope that the Portiloop will increase research on closed-loop stimulation, and continue to evolve and develop as a community-supported tool.

## 2 General background

### 2.1 Limitations of current systems and design objectives

Speed, expense, flexibility, and portability are important considerations for designing a highly functional research-focused closed-loop system. The brain's endogenous oscillations range from about 0.1 to 150 Hz. Depending on the application and the neural event of interest, real-time constraints can vary from a few ms [16] to seconds [4]. Currently available commercial systems that are capable of slow oscillation closed-loop stimulation have difficulty accurately and precisely detecting and stimulating faster, higher frequency neural events. Devices that are fast enough and flexible enough for research purposes tend to be derived from high-end systems used for real-time computing in other applications, e.g., in aerospace and automotive industry [1], and are large and expensive.

Various portable devices have been developed to acquire and process EEG signals. In McCrimmon et al. [17], the authors developed a low-cost device limited to acquisition. Other portable devices enable closed-loop stimulation [2–4], some also based on low-cost hardware [9], but work with simple heuristics and are generally not sufficiently powerful for complex signal detection algorithms such as those based on deep learning. Our goal is to design a closed-loop system that runs on inexpensive, portable hardware, yet is still sufficiently fast, powerful, and flexible for cutting-edge research. Another element of experimental flexibility that we incorporate into the design is the capability to change the input and output signals. Thus, although our current focus is EEG and auditory stimulation, an EEG trace could be exchanged for another physiological signal like that derived from functional near infrared spectroscopy, and detection output could be used to stimulate the brain more forcefully using transcranial electrical or magnetic stimulation. By designing the system flexibly such that it can be extended to detect and stimulate a variety of brain oscillations, we can greatly expand its application, for example to theta-band oscillations that are associated with working memory capacity and task performance [18], or sleep spindles. The Portiloop is designed to be the first system to provide a portable, real-time and deep learning-capable solution for multiple fundamental research applications.

## 2.2 Sleep spindles as a challenging test case

Slow oscillations, which have been the main target for closed-loop auditory stimulation (CLAS) to date, are thought to work in concert with other faster oscillations, called *sleep spindles*, to reactivate recently learned memories and transfer them to long-term memory [19, 20]. Sleep spindles are transient oscillations observed in both lighter and deeper non-rapid eye movement (NREM) sleep (*i.e.*, sleep stages 2 and 3). Their role in memory consolidation is supported by increases in spindle density following learning (*e.g.*, [21]), and the observation that age-related changes in sleep spindles are correlated with differences in overnight performance gains (*e.g.*, [22, 23]; see [24] for a review of spindle mechanisms and functions).

If it were possible to influence spindles with sound, as it is to enhance slow oscillations, researchers could explore their functional role in healthy adults as well as characterize their involvement in cognitive aging, and even perhaps restore degraded function. Particular challenges of spindle stimulation are that each oscillatory cycle is only ∼60 ms long and the entire spindle is between 0.5 and 2.5 s, leaving little time for traditional window-based frequency analysis; there is considerable variability between the frequency, amplitude, and duration of individuals' spindles, particularly in older populations [25, 26]; and even for offline detection of spindles (which is an easier task than detecting spindles online, as the entire spindle is available and can be used in detection), agreement on spindle identification between experts themselves is limited (∼70%) [27, 28]. Real-time detection of spindles is therefore a challenging test case for the Portiloop, and a working online spindle detector would be of direct interest as a research tool.

## 2.3 Offline sleep spindle detection for labeling and performance comparison

Machine learning-based detection algorithms are powerful means of detecting subtle signals in physiological data such as EEG, but they require large sets of accurately labeled data for training and testing the algorithm's performance. Once trained, the success of an algorithm on classifying previously unseen data can be quantified using the f1-score, which is a widely used metric to quantify an average of recall (i.e., success in detecting events) and precision (i.e., the proportion of detected events that are correct), see S1 File for equations. The consistent detection and labeling of sleep spindles is a challenging task, due to variability in their appearance and strength. Traditionally, spindles have been visually identified by multiple experts, with f1-scores computed for each scorer with respect to spindles identified by the consensus. One commonly used dataset for creating and testing spindle detection algorithms [29, 30] is the Montreal Archive of Sleep Studies (MASS) [31], in which the sleep spindle annotations were provided by two experts. Projects using MASS for training usually take spindles identified by either expert (i.e., a logical "OR" operation). However, the MASS annotations have a low inter-rater agreement (f1-score = 0.54 [28]), which makes this procedure statistically naive. The Massive Online Data Annotation (MODA) [28] project addressed this issue by having 5 experts (on average) annotate spindles on a subset of data from MASS, and rate their confidence, in each EEG segment. The experts had an inter-rater f1-score of 0.72 with respect to the final MODA labels. This score is considerably better than the MASS equivalent, and the number of experts, the scoring and the post-processing steps enable final labels of much higher precision. We therefore adopt MODA as a basis for performance measurement, bearing in mind that even MODA does not provide a true answer about whether a spindle has occurred or not; only some degree of consensus.

Several offline sleep spindle detectors have been developed and tested on MODA [27, 32– 37]. However, these generally use heuristics that compute Fourier transforms or wavelet

decomposition on large portions of the signal. For real-time detection in online applications, spindles must be detected soon after their onset, if stimulation is to arrive before the spindle ends and thus be capable of influencing its evolution. Online real-time detectors therefore cannot take the same approaches that have been successful for offline detection.

## 2.4 Considerations for online sleep spindle detection

Online detectors (*i.e.*, detectors that act during signal acquisition) face more challenging conditions than offline detectors, due to the unavailability of "future" data points. For example, if we aim to detect and stimulate a spindle before it ends, the duration of the spindle is not yet known by definition, yet it is one of the identifying criteria for spindles commonly used by experts. Some existing heuristics filter the signal, compute power features and rely on thresholds to perform detection; however, these approaches yield relatively poor f1-scores [9].

Deep learning can also be leveraged to perform online sleep spindle detection. This is done by first training an artificial neural network offline through supervised learning to detect sleep spindles, and then feeding the incoming signal to the trained detector. Several such models have been trained in previous work [29, 30, 38, 39]. However, these works do not consider hardware constraints that are central for our purpose: they use large models that are often unable to run in real time even on high-end GPUs, which makes them inapplicable in embedded systems. Moreover, they are usually trained and tested on MASS [31] with an "OR" operation performed on the two experts' labels, which as discussed above is not a highly precise target [28].

In this work, we design a Pareto-optimal neural architecture that performs best on the MODA dataset [28] while satisfying our hardware and timing constraints. We validate our architecture against the state-of-the-art SpindleNet [29], initially used with the MASS dataset. When both architectures are trained and tested on MODA, ours vastly outperforms the baseline, on top of running in real time on embedded hardware.

## 3 The Portiloop system

A high-level description of the Portiloop system is provided in Fig 1(a), while a more detailed implementation scheme can be found in Fig 1(b). Fundamentally, it is made of an EEG front-end connected to an embedded computer which reads the EEG signals, filters them, feeds the filtered signal to an Artificial Neural Network (ANN) trained to detect specific signals, and generates a stimulus when a target pattern is detected.

We propose two implementations of the Portiloop that can be replicated by readers with the appropriate technical background:

- A version that can be fully built using off-the-shelf components based on a Xilinx Pynq FPGA board and an 8-channel HackEEG frontend (Fig 1(c))

- A custom printed circuit board (PCB) featuring an EEG frontend and a Google Coral neural accelerator (Fig 1(d))

The detailed hardware implementation is out of the scope of this paper, but readers can find all instructions and plans in our open-source repository.

Since closed-loop stimulation requires very precise timing, the Portiloop needs to detect target pattern as quickly as possible, and minimize the delay of the output stimulus. We identify two different sources of delay in the proposed system, *hardware* and *software* delays. By *hardware delays* we refer to the time it takes to retrieve the signal from the electrodes, convert it to digital, filter it, process it through the ANN, and send the resulting feedback stimulation to the subject. By *software delays* we refer to time required for our system to collect enough

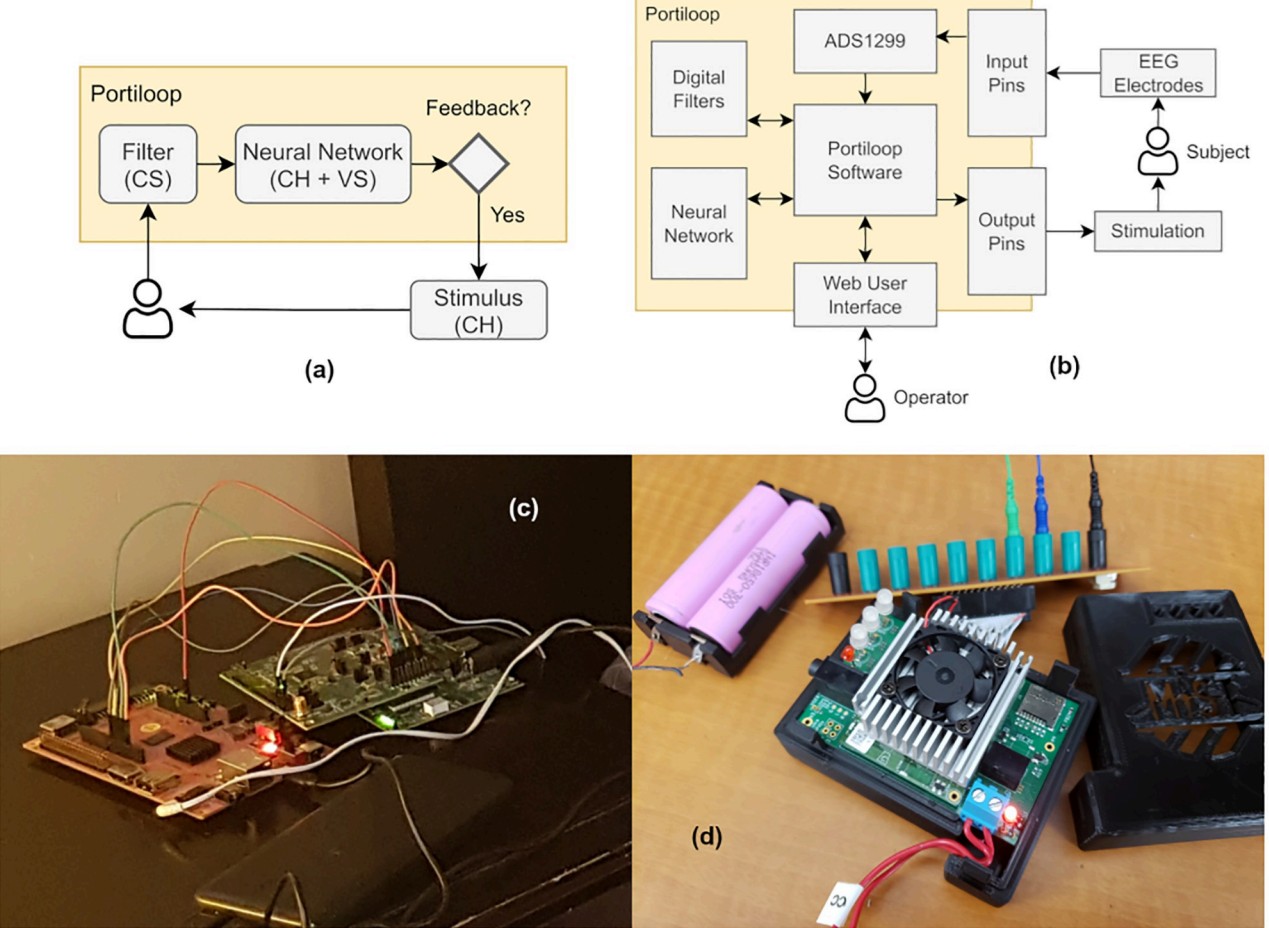

**Fig 1. Portiloop implementation.** (a) High-level view of the system: captured EEG data is first filtered and fed into a neural network. If a neural event is detected with sufficient confidence (>threshold), a decision is made to initiate brain stimulation. The types of delays introduced by each component are denoted in parenthesis: C/V denote Constant/Variable delays, and H/S denote Hardware/Software delays. (b) A detailed implementation scheme, and two possible implementations: (c) an FPGA prototype based on off-the-shelf components, and (d) a Coral-based implementation that uses a custom printed circuit board. Plans are available on our GitHub page.

data to perform its functions. As an example, although the hardware operations performed by signal filters are near-instantaneous, filtering requires that a certain amount of data be collected before outputting a filtered value, introducing a constant software delay in the output signal. This delay is a trade-off related to the order of the filter. The higher the order of a filter, the more efficient it is at removing undesirable frequencies, but also the longer the software delay introduced in the signal by the filtering operation. Similarly, an ANN may need to "see" a certain portion of a signal to recognize it, introducing a (generally variable) delay on the output of the classifier. An example of such delay is illustrated in Fig 2, where the trained ANN that we latter describe in Section 6 takes a variable amount of time before correctly detecting a transient pattern in EEG signal. These hardware and software delays sum to a total delay that is the response time of the Portiloop system. They depend on the target signal and put limits on the timing constraints of the application.

The Portiloop GitHub includes a software for recording and visualizing the EEG signal on the device, as well as Python programming interface for the development of extensions or new

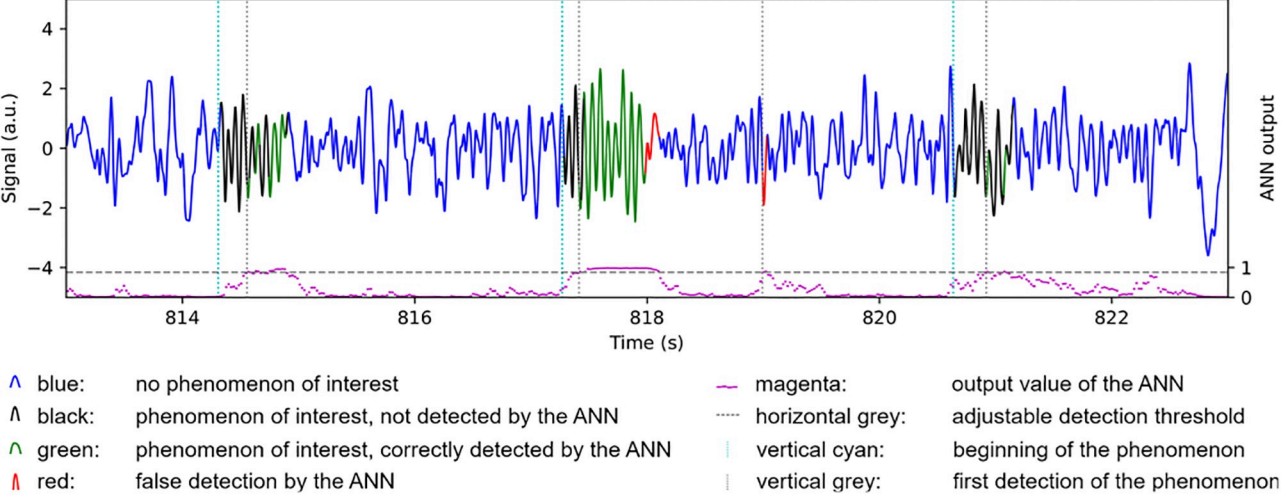

**Fig 2. Real-time stimulation example of the Portiloop on sleep spindles.** The output of the ANN (likelihood between 0 and 1, magenta) is displayed in the lower part of the Figure. When it crosses an adjustable detection threshold (horizontal grey line, here set to 0.84), the Portiloop sends a stimulus (vertical grey line). The optimal target for this stimulus is the beginning of the sleep spindle (vertical cyan line). Thus, the variable software delay is visible here between the vertical cyan and grey lines. The sections of the signal in red mark are sections wrongly detected as spindles (false positives), and the areas in black those that are not, or not yet, detected as spindles but were identified as spindles by experts (false negatives).

applications. The Portiloop can be accessed via WiFi. A web-based Graphical User Interface (GUI) allows to configure the EEG channels, set up detection and stimulation, visualize the signals in real-time, record EEG, set up custom filters, and more. The recording can be saved either in the internal memory (32GB) or an SD card in EDF format, or streamed through the network using the Lab Streaming Layer (LSL) [https://github.com/sccn/labstreaminglayer], which timestamps the data stream with microsecond accuracy.

## 4 Neural network implementation

The Portiloop is primarily designed for EEG signals, i.e. time-series of data containing oscillatory and transitory elements. In the realm of deep learning, a natural way of processing such data is to use either 1D convolutions, recurrent units, or a combination of both. The type of ANN architecture that we recommend is inspired by SpindleNet [29]. In essence, a sliding window over a few last data points is fed to a Convolutional Neural Network (CNN) whose purpose is to extract relevant features (*e.g.*, frequencies) in this signal fragment. Then, these extracted features are fed to a Recurrent Neural Network (RNN) whose purpose is to keep track of the features extracted in past forward passes (where a "forward pass" is the action of computing an output from the ANN). Note that another family of architectures, called Transformers [40], is known for exhibiting good results with this type of data when infinite compute is available for inference. However, Transformers are memory-less and not suitable for lightweight real-time applications, because they need to process the whole signal at each forward pass. Conversely, RNNs are able to process one single data point at each forward pass and keep track of the past in memory, which makes them more applicable for the Portiloop.

The Portiloop has a limited amount of available memory, so as to ensure its portability and low price. Therefore, large ANN architectures such as SpindleNet [29] are orders of magnitude too large to be implemented in our device. To produce networks that are suitable to our device, we rely on an automated optimization algorithm named "Parallel Model-Based Optimization"

(PMBO) that allows us to trade-off accuracy and use of resources on our device (see S1 File for details).

In addition, given the Portiloop's design constraints, we sought a lightweight means of allowing our resource-restricted network to use as much signal history as possible (as do larger neural networks). Time dilation [41] is a technique that enables recurrent units such as Gated Recurrent Units (GRUs) to look further back in time before gradients vanish, at no computational cost. In the S1 File, we propose a version of this technique that allows us to virtually parallelize a single physical ANN into several decoupled virtual models. Our approach enables shallow recurrent neural networks to look further back in time by skipping the redundant information that is inherent to the use of a sliding window as input, while still acting as fast as possible.

## 5 Case study: Online sleep spindle detection

We now turn our attention to a case study application of the Portiloop in neuroscience—detecting sleep spindles shortly after they start so as to be able to stimulate the brain during the spindle. The long-term goal of this application is to further clarify the role of sleep spindles in learning and memory, and to explore therapeutic interventions for memory decline (see Section 2.2). As described in Section 2.4, stimulating sleep spindles is a particularly challenging case study due to their high frequency ($\sim$ 12 to 16 Hz) and rapid evolution ($<$2.5 s), and therefore tight timing constraints, and thus serves as a demonstration of the technology's capabilities.

To the best of our knowledge, the state-of-the-art in previous work regarding online sleep spindles detection was SpindleNet [29]. This architecture has too many parameters to be implemented on anything but the largest graphics processing units. Moreover, it was trained and evaluated on the MASS labels (i.e., a logic "OR" on the annotations of two experts whose spindle evaluation varies considerably). Since we do not have access to the SpindleNet model, which is closed-source, we rebuilt the architecture described in [29] and trained it on the more difficult MODA dataset [28] with the same pipeline that we used to train our models, as a means of comparing the models' performance.

We draw inspiration from SpindleNet as a starting point for our ANN architecture design. In particular, we train models based on the same idea of using Convolutional Neural Networks (CNNs) followed by Recurrent Neural Networks (RNNs), and we evaluate the relevance of the three different inputs used by SpindleNet (namely, the raw signal, the signal envelope and the signal's power features) in our setting. We then use our optimization algorithm (named PMBO) along with the MODA dataset to derive a much smaller architecture, and provide a quantitative comparison with the SpindleNet architecture on MODA. Since maximum experimental flexibility is attained by being able to stimulate anytime during the course of the spindle including with phase precision, we conduct a thorough time analysis of the proposed system, and document possible trade-offs that a researcher might use to maximize performance for a given experimental application.

### 5.1 Dataset and training

We use the MODA dataset (a subset of MASS), for training our ANN, since its labels are considerably more reliable [28]. Ethical approval for use of the dataset was obtained from the database's scientific committee and Concordia University's Research Ethics Unit. This dataset is divided in two subsets. The first one, called *phase 1*, consists of 100 younger subjects, whereas the second one, *phase 2*, consists of 80 older subjects. The MODA dataset provides two types of annotations (labels) on the signal: the first is the mean score given by the group of experts

for each data point; the second is a binary classification of each data point as a spindle or non-spindle, defined by a threshold on the aforementioned scores (0.2 for phase 1 and 0.35 for phase 2). Further post-processing steps were applied to obtain these binary labels: spindles that were too short (<0.3 s) and too close (<0.1 s) to each other were merged, then spindles that were too short (<0.3 s) or too long (>2.5 s) were relabeled as negative. Given this dataset, two types of ANNs are possible: classifiers and regressors. These two types of ANNs differ only by the labels and losses used to train them. Classifiers are trained on the binary labels, by optimizing the binary cross entropy loss. They directly predict whether the current signal is a spindle or not, according to the very specific definition given by these binary labels (*i.e.*, taking into account the thresholds and post-processing applied by MODA). Regressors are trained on the score labels, by optimizing the mean square error loss. They predict the score given by the experts (before the aforementioned post-processing steps), which allows the user to select their own threshold for detection. Note that, in practice, classifiers also enable the user to select their own threshold, although in a less interpretable way. We experiment with both types of models. Finally, note that MODA is a highly unbalanced dataset as only about 5% of the signal is labeled as sleep spindles. During the course of this work, we tried different ways of balancing training for classifiers and regressors. Interestingly, we found that classifiers benefit highly from oversampling (*i.e.*, sampling 50% of spindles and 50% of non-spindles from the dataset during training) whereas all the balancing techniques we tried for regression (including oversampling, Label Distribution Smoothing [42] and a custom version of the latter) actually hinder training.

To evaluate against SpindleNet we compute the inputs used by this model: the signal, the envelope of the signal, and a "power feature ratio" [29]. The latter compares frequencies between 2 Hz and 8 Hz with frequencies between 9 Hz and 16 Hz from the Fourier transform over the last 500 ms of signal. Computing this ratio is resource-intensive in the context of the Portiloop system, and furthermore did not improve our models' performance. Therefore, we compute this ratio offline for the sole purpose of comparison with SpindleNet, and we do not use it in our model. We set the sampling frequency to 500 Hz, which allows the Portiloop to log the raw signal at a higher resolution, and then downsample to 250 Hz. Fig 3 depicts the pipeline that computes the cleaned signal and envelope.

We filter the EEG signal in the same frequency band as used in standard sleep scoring (*i.e.*, 0.5 Hz to 30 Hz) [43]. An FIR filter of order 20 works reasonably well to remove frequencies above 30 Hz, but we observed persistent power line noise in unshielded home or office recording environments. To address this issue, we apply a notch filter whose frequency depends on the geographical area (50 Hz in Europe, 60 Hz in North America). For removing low frequencies, we rely on online standardization through exponential moving average (formulas are

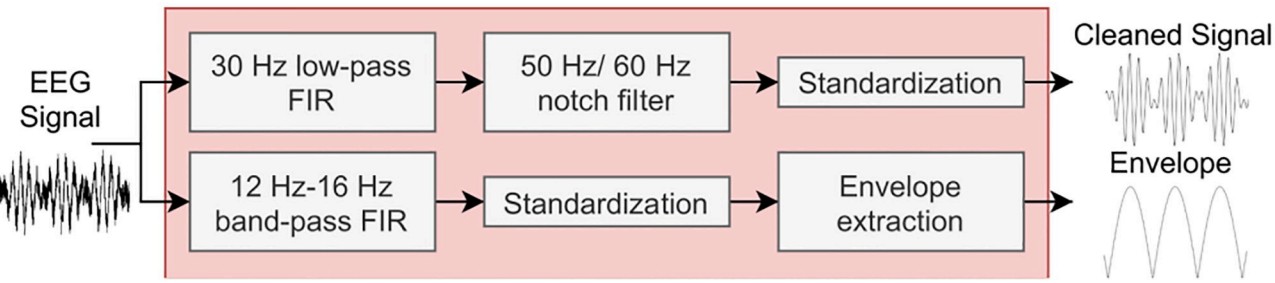

**Fig 3. Portiloop signal processing pipeline for extracting relevant inputs for the ANN.** The selected filters introduce an identical software delay of 40 ms in both branches. (Note that power features are computed offline only for the SpindleNet architecture and are not represented in this diagram).

provided in S1 File). We use a coefficient $\alpha_\mu = 0.1$ for our running average, which attenuates frequencies under 4 Hz. We use a smaller coefficient $\alpha_\sigma = 0.001$ for our running variance, meaning that our estimate of the standard deviation takes a larger portion of the signal into account. We empirically found this choice of $\alpha_\mu$ and $\alpha_\sigma$ to reveal EEG features of interest and yield acceptable standardization, by visual inspection.

We apply a similar procedure to extract the envelope: first, we filter the signal with a FIR band-pass between 12 Hz and 16 Hz. Then, we standardize with $\alpha_\mu = \alpha_\sigma = 0.001$, we square the signal, and we smooth the result by computing its moving average, this time with $\alpha_\mu = 0.01$. We evaluate different types of ANN architectures, using either both or only one of these pre-processed signals as input. Since FIR filters introduce software delays, we have designed both branches of the pipeline so that they introduce identical software delays to their respective outputs (*i.e.*, 40 ms at 250 Hz sampling rate with FIR filters of order 20).

The output of the ANN tells whether the model considers the current signal being a sleep spindle or not. Some further processing is necessary to ensure that we only send one stimulation per spindle. As seen in Fig 2, the detection can be noisy around the beginning or the end of a spindle, especially since we use decoupled virtual parallel networks (see S1 File). A stimulus is sent upon initial spindle detection. To avoid multiple stimuli of the same spindle, the subsequent stimulation may only occur 400 ms following the end of the spindle. If a spindle is detected again within this duration the timer is reset, and we consider it as being part of the previous spindle.

## 6 Validation and performance

We report results from a thorough quantitative and qualitative study of the system, not only in terms of detection scores as generally seen in previous work (i.e, proportion of data points correctly detected to be part of a spindle), but also in terms of real-time stimulation performance. Note that all our experiments are based on the MODA dataset rather than actual nights spent wearing the Portiloop device, as we would not have ground truth labels for newly recorded data. Further validation of the final device will require reproducing the experimental setting of MASS/MODA with the Portiloop (while participants are simultaneously wearing a research-grade polysomnography system for comparison) and labelling acquired data.

All results regarding online detection performance are summarized in Table 1. This table shows the f1, precision and recall metrics that statistically describe how efficient different models are at detecting sleep spindles (on average over all data points). These metrics are provided separately for phase 1, which groups younger subjects, for phase 2, which groups older subjects, and for the whole cohort.

As previously highlighted, sleep spindle detection is a difficult task and experts themselves often do not agree when annotating these offline. This disagreement is quantified by MODA [28] and represented in Table 1, row (1) for reference. The experts annotating the MODA dataset had an average performance of 0.72 on the whole cohort in term of the f1-score of their individual annotations with respect to the final labels. They are compared to other *offline detection*, *i.e.*, when a virtually infinite computational budget and the whole signal is available, including future data points, presented under "offline detection" in Table 1 (taken from [28]). We instead perform *online detection*, which has additional challenges: (a) computation happens in real time; (b) the future signal is not available.

The MODA dataset is relatively small ($\sim 24$ h of annotated data) and heterogeneous. This adds some difficulty for training and properly assessing the performance of our models, because we choose to use only 10% of subjects as our validation set (for model selection), and another 10% of subjects as our test set (for final model evaluation). Since the results would

**Table 1. Quantitative results. Our different models and ablations are compared under "Online Detection" using the nomenclature "mean (std)", and superscripts for referencing rows in the text.** In rows (4) and (5) we replace an input of our 2-input model by a copy of the other, in row (7) we remove time-dilation, in rows (8) and (9) we train our model only on phase 1 or phase 2 (i.e., young subjects or old subjects), and in row (10) we train a regressor to evaluate it as a classifier.

| | (a) Phase 1 (younger) | | | (b) Phase 2 (older) | | | (c) Whole Cohort | | |
|---|---|---|---|---|---|---|---|---|---|
| | *Recall* | *Precision* | *f1* | *Recall* | *Precision* | *f1* | *Recall* | *Precision* | *f1* |
| | Experts | | | | | | | | |
| *Inter-rater agreement[1]* | 0.76 (0.16) | 0.81 (0.17) | **0.76 (0.1)** | 0.66 (0.19) | 0.74 (0.17) | **0.65 (0.12)** | 0.72 (0.18) | 0.78 (0.17) | **0.72 (0.12)** |
| | Offline Detection | | | | | | | | |
| *Ferrarelli [32]* | 0.19 | 0.83 | 0.31 | 0.16 | 0.87 | 0.27 | 0.18 | 0.85 | 0.29 |
| *Mölle [33]* | 0.83 | 0.47 | 0.6 | 0.78 | 0.44 | 0.56 | 0.81 | 0.46 | 0.58 |
| *Martin [34]* | 0.61 | 0.64 | 0.62 | 0.58 | 0.56 | 0.57 | 0.6 | 0.6 | 0.6 |
| *Wamsley [35]* | 0.57 | 0.69 | 0.63 | 0.56 | 0.62 | 0.59 | 0.57 | 0.66 | 0.61 |
| *Lacourse [27]* | 0.75 | 0.73 | **0.74** | 0.7 | 0.69 | 0.7 | 0.73 | 0.71 | **0.72** |
| *Ray [36]* | 0.73 | 0.47 | 0.57 | 0.75 | 0.32 | 0.45 | 0.74 | 0.4 | 0.51 |
| *Parekh [37]* | 0.85 | 0.61 | 0.71 | 0.74 | 0.68 | **0.71** | 0.8 | 0.65 | 0.71 |
| | Online Detection | | | | | | | | |
| *Based on SpindleNet [29][2]* | 0.92 (0.04) | 0.24 (0.07) | 0.38 (0.07) | 0.85 (0.06) | 0.19 (0.08) | 0.3 (0.1) | 0.89 (0.05) | 0.22 (0.07) | 0.35 (0.08) |
| **2-input[3]** | 0.68 (0.04) | 0.6 (0.06) | **0.64 (0.03)** | 0.52 (0.09) | 0.58 (0.04) | 0.54 (0.05) | 0.62 (0.06) | 0.6 (0.05) | **0.61 (0.03)** |
| **2-input ablation 1[4]** | 0.7 (0.09) | 0.47 (0.08) | 0.55 (0.04) | 0.56 (0.11) | 0.43 (0.09) | 0.47 (0.04) | 0.65 (0.1) | 0.46 (0.08) | 0.52 (0.04) |
| **2-input ablation 2[5]** | 0.72 (0.03) | 0.57 (0.06) | **0.64 (0.03)** | 0.57 (0.08) | 0.53 (0.04) | **0.55 (0.04)** | 0.67 (0.04) | 0.56 (0.05) | **0.61 (0.03)** |
| **1-input[6]** | 0.7 (0.04) | 0.59 (0.05) | **0.64 (0.03)** | 0.54 (0.09) | 0.58 (0.05) | **0.55 (0.05)** | 0.64 (0.05) | 0.59 (0.05) | **0.61 (0.03)** |
| **1-input ablation td[7]** | 0.47 (0.1) | 0.6 (0.09) | 0.51 (0.03) | 0.31 (0.12) | 0.59 (0.08) | 0.39 (0.08) | 0.41 (0.1) | 0.6 (0.09) | 0.47 (0.04) |
| **1-input trained on p1[8]** | 0.72 (0.05) | 0.56 (0.05) | 0.63 (0.03) | 0.57 (0.08) | 0.52 (0.07) | 0.54 (0.05) | 0.66 (0.07) | 0.55 (0.05) | 0.6 (0.03) |
| **1-input trained on p2[9]** | 0.75 (0.05) | 0.5 (0.05) | 0.6 (0.02) | 0.62 (0.09) | 0.45 (0.05) | 0.52 (0.03) | 0.7 (0.06) | 0.49 (0.04) | 0.57 (0.02) |
| **1-input regression[10]** | 0.62 (0.07) | 0.64 (0.06) | 0.63 (0.03) | 0.53 (0.06) | 0.55 (0.08) | 0.53 (0.04) | 0.58 (0.06) | 0.62 (0.06) | 0.6 (0.03) |

otherwise be dependent on the assignment of subjects to the three sets, we evaluate our models through the following procedure:

- we shuffle all subjects 10 times and compute a different training/validation/test split of the dataset each time (sets are thus made of separate subjects);

- for each split, we use the training set to train 3 models, the validation set being used to estimate their f1-score. We select the best of these 3 models by its best f1-score on the validation set. We then report the performance of this model in terms of its f1-score on the test set;

- the above being repeated 10 times, we report the average test f1-score in Table 1, with the corresponding standard deviation being indicated in parenthesis.

As described previously, we use the SpindleNet [29] architecture as a baseline for evaluating the performance of our own models. Since SpindleNet is closed-source and trained on the MASS dataset, we retrain its architecture from scratch with the same pipeline as used to train our other classifiers. In particular, we balance training through oversampling (as opposed to the data augmentation technique used by the authors of the original paper), and we train and evaluate SpindleNet on the MODA dataset. The results of this experiment are presented in Table 1, row (2). The baseline has a high recall and a poor precision; in other words it it tends to incorrectly label non-spindle events as spindles.

We first derive a lightweight ANN architecture by drawing inspiration from SpindleNet. More precisely, we use our optimization algorithm PMBO to find a Pareto-optimal architecture that uses both the cleaned signal and the envelope as inputs. The resulting architecture is presented in the Supplementary Information. We measure a total duration of 40 ms for each

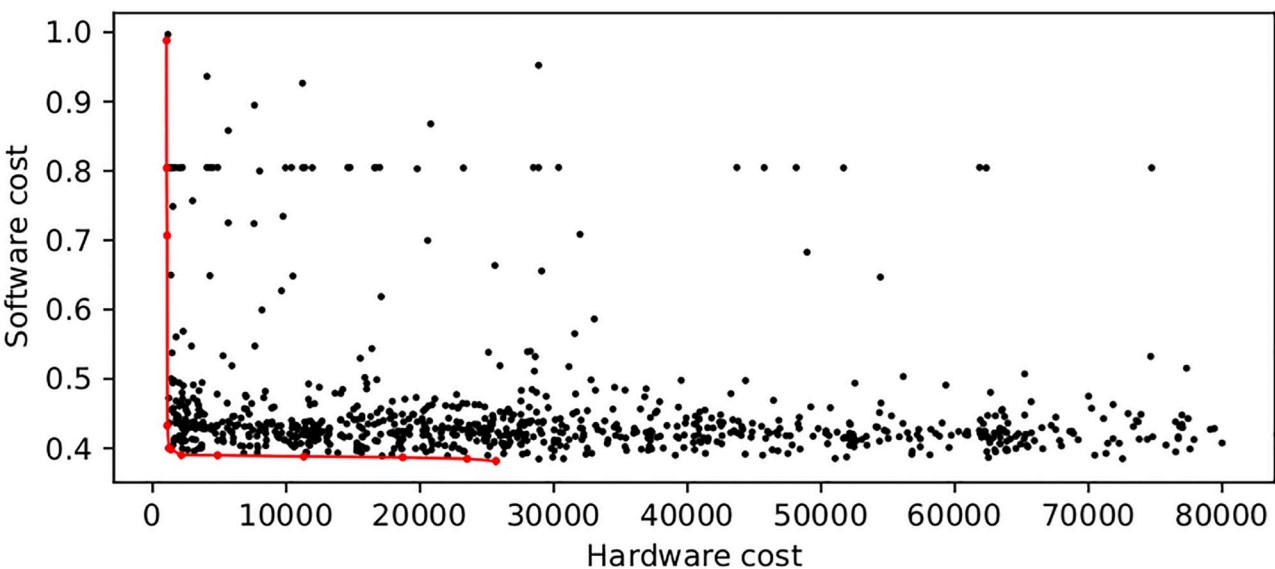

**Fig 4. Search space of the single-input architecture, found with PMBO.** The hardware cost is the number of trainable parameters in the neural architecture, and the software cost is 1−f1- score of the fully- trained model. Black dots: non-Pareto-optimal models tested by the algorithm. Red dots: Pareto-optimal models found by the algorithm. Red line: Pareto front. The researcher would select a configuration from the Pareto front, which represents optimal trade-offs between both costs.

forward pass in this model on the FPGA-based variant of the Portiloop. The detection performance of this model, reported in Table 1, row (3), vastly outperforms the baseline.

The idea of using the envelope along the raw signal as input to the ANN is drawn from the baseline. Since the envelope is computed from the raw signal, it should not contain any additional information that cannot be extracted by an ANN. To evaluate the relevance of this particular input, we perform the following ablation: to keep the same architecture (and thus the same model capacity), we replace one of the two inputs by a copy of the other. In Table 1, row (4) both inputs are the envelope, while in row (5) both inputs are the cleaned signal. We find that the envelope input can be removed: the model in which we replace the envelope with a copy of the cleaned signal (5) has the same performance as the original model (3), and even performs marginally better on phase 2.

Since we deem the use of the envelope input ineffective, we use PMBO one more time to devise our final Pareto-optimal ANN architecture, now with only the cleaned signal as input. For this matter, we run PMBO on 20 Tesla V100 GPU workers over a period of 24h. The detailed hyperparameters used in this experiment are provided in S1 File, and the results are visualized in Fig 4, which shows all the explored architectures according to their classification performance (software cost) and the use of FPGA resources (hardware cost). The red line is the Pareto front, meaning the set of configurations that are optimal for at least one of the two metrics: this means all points that are *not* on the Pareto front have at least one corresponding configuration that is better in terms of both software and hardware cost, and should therefore not be considered. We select the best model in terms of software cost (i.e. the one with the highest classification performance) irrespective of its hardware cost *i.e.*, the model corresponding to the right-hand end of the Pareto front. This model is acceptable because it is anyway rather small, with only 25.6k parameters. We measure the execution time of this architecture to be 20 ms per forward pass on the Portiloop (vs. 40 ms for the 2-input version).

The selected architecture is described in Fig 5, and its detection performance is summarized in Table 1, row (6). Compared to our 2-input model, the single-input model exhibits the same performance, with even a marginal improvement on phase 2, while executing twice as fast (20 ms versus 40 ms). The detailed hyperparameters of this model are provided in S1 File.

To verify that the use of virtual parallelization via time-dilation (c.f. S1 File) is indeed necessary to obtain our results, we shrink the time-dilation (set to 168 ms by PMBO) to the minimum, *i.e.*, 20 ms since this is the execution duration of the ANN per forward pass. This removes the virtual parallelization, since the same ANN must now be used for each sample. Therefore, each step of back-propagation reaches 8 times less far back in time during training. The result of this ablation is presented in Table 1, row (7). The highly deteriorated results illustrate the importance of time-dilation. This hints at the relevance of looking relatively far back in time to annotate sleep spindles.

Finally, to ensure the generality of our ANN, and knowing that spindles change in older adults [24], we compare the results using the data of MODA phase 1 (younger subjects) and the data of MODA phase 2 (older subjects). Namely, we either train the model on subjects drawn only from phase 1, or subjects drawn only from phase 2. The results of these experiments are presented in Table 1, rows (8, 9). We observe that the ANN trained on phase 1 performs almost as well as the ANN trained on the whole cohort (6) on all subsets, including phase 2, whereas the ANN trained on phase 2 is noticeably worse on all subsets, even including phase 2. We hypothesize that this is because phase 2 is comprised of older adults, who have lower amplitude and fewer sleep spindles. Using phase 2 during training is still useful in terms of generalization. Indeed, the ANN trained on phase 1 only (8) has a slightly worse performance when tested on phase 1 than the ANN trained on the whole cohort (6).

Note that all models presented beforehand are classifiers. We also train a regressor with the same architecture, as explained in Section 5.1. There is a subtle difference in what this model measures when compared to our classifiers: whereas classifiers predict whether the signal is a sleep spindle according to the full definition given by MODA (including post-processing), the regressor predicts the mean score given by the experts (excluding post-processing). Since we are primarily interested in classification in this article, we find the threshold that maximizes the f1-score on the binary labels, presented in S1 File. We find that the optimal threshold is 0.27 for phase 1, 0.23 for phase 2 and 0.26 for the whole cohort. We then evaluate the regressor with these thresholds on the classification task and report the results in Table 1, row (10). These results are slightly weaker than those of the classifier (6). We surmise that this effect comes from the post-processing steps performed by MODA to compute the binary labels. We choose the 1-input classifier (6) for the remainder of this article.

## 6.1 Real-time stimulation

The performance measured in the previous section is not entirely representative of the performance on the final task. So far, we have only measured the capability of the model to annotate each data point of the signal individually. Yet, we want the ability to send one single stimulation per sleep spindle.

The ANN delays must be compounded with the other sources of delays (here reported for the FPGA version as a worst case, as they are slightly lower for the Coral version), *i.e.*, the software delay from FIRs (40 ms), the ANN forward pass duration (20 ms) and the stimulation hardware delay, to measure our real stimulation performance. We measure an auditory stimulation delay of 4 ms when using a basic sound controller, for a total constant delay of 64 ms. The measured delays are summarized in Table 2, were one can see that the most significant

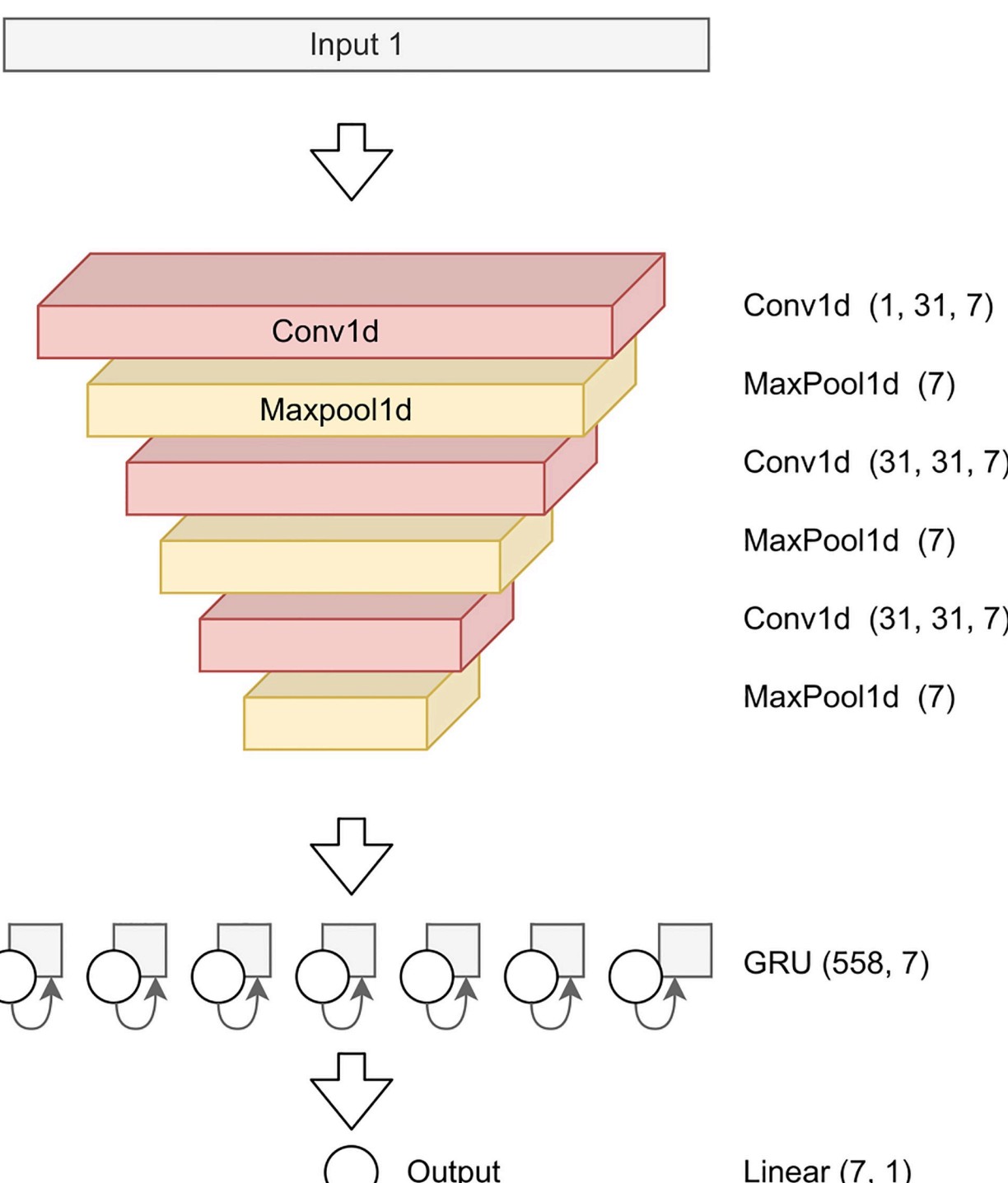

**Fig 5. Final single-input ANN architecture.** The dimensions of each layer are provided in parenthesis using the PyTorch nomenclature.

**Table 2. Delays measured in the Portiloop (sleep spindle configuration).**

| Component | Hardware delay | Software delay |
|---|---|---|
| Electrodes + ADC | - | - |
| Filters | - | 40 ms |
| ANN | 20 ms | $\sim 250\ (\pm 100)$ ms |
| Stimulus | 4 ms | - |
| Total: $\sim 314$ ms | 24 ms | $290\ (\pm 100)$ ms |

source of delay is the detection delay of our ANN. Training a faster model is thus a potential avenue for future work.

From now on, we redefine: (a) True positive: the first stimulus sent within the duration of a spindle, taking all delays into account; (b) False positive: any other stimulus; (c) False negative: any spindle that does not receive a stimulus within its labeled duration.

Fig 6(a) displays the detection performance (taking all delays into account) of our final device. We compute the stimulation precision, recall and f1-score according to the aforementioned definitions of true positives, false positives and false negatives. This provides a visualization of possible trade-offs in terms of how many spindles we want to stimulate (recall) versus how sure we want to be that all stimuli are relevant (precision). In terms of f1-score, the best such trade-off is attained at a threshold of 0.84 with our model, yielding a precision and a recall of 0.71 both.

The timing performance of our system can be observed in Fig 6(b), which displays the distribution of stimulation delays, *i.e.*, the distribution of the stimulus being closest to the beginning of each sleep spindle, all delays being taken into account. Some stimulation delays are negative, as spindles are sometimes stimulated in advance (note that we count these as false positives, which slightly harms our reported results). Fig 6(b) shows the effect of increasing the detection threshold of our model on the stimulation delays. According to Fig 6(a), choosing a 0.84 detection threshold over the 0.5 default classification threshold in our ANN yields a better stimulation f1-score and in particular much more precise stimuli, but this comes at the price of slightly shifting the stimulation delay distribution to the right, *i.e.*, introducing some additional delay to the stimulation, as further seen in S1 File.

To further illustrate the final performance of the system, Fig 2 displays an example of its real-time stimulation capability on actual EEG signal (test dataset). More examples and visual insights are provided in S1 File.

Finally, we estimate the Portiloop energy efficiency by running the FPGA version continuously, powered by a fully-charged 20000 mAh battery. The battery dies out after 26 hours and 22 minutes, suggesting that our power consumption is roughly 756 mA. The Coral version runs for approximately 8 hours with a 12000 mAh battery pack suggesting a 1500 mA current draw.

## 7 Discussion and future work

In this article, we introduce the Portiloop, a device that enables the real-time detection and stimulation of patterns of interest in electroencephalography signals. Our system is open-source, portable, low-cost, and can be tailored for many brain stimulation research applications. We propose a pipeline to design neural architectures that are relevant for processing EEG signals in real time. We further propose an algorithm that automates the process of finding efficient models (i.e., PMBO), using one-to-many parallel workers. We demonstrate our proposed system on the closed-loop stimulation of sleep spindles, a difficult task of high

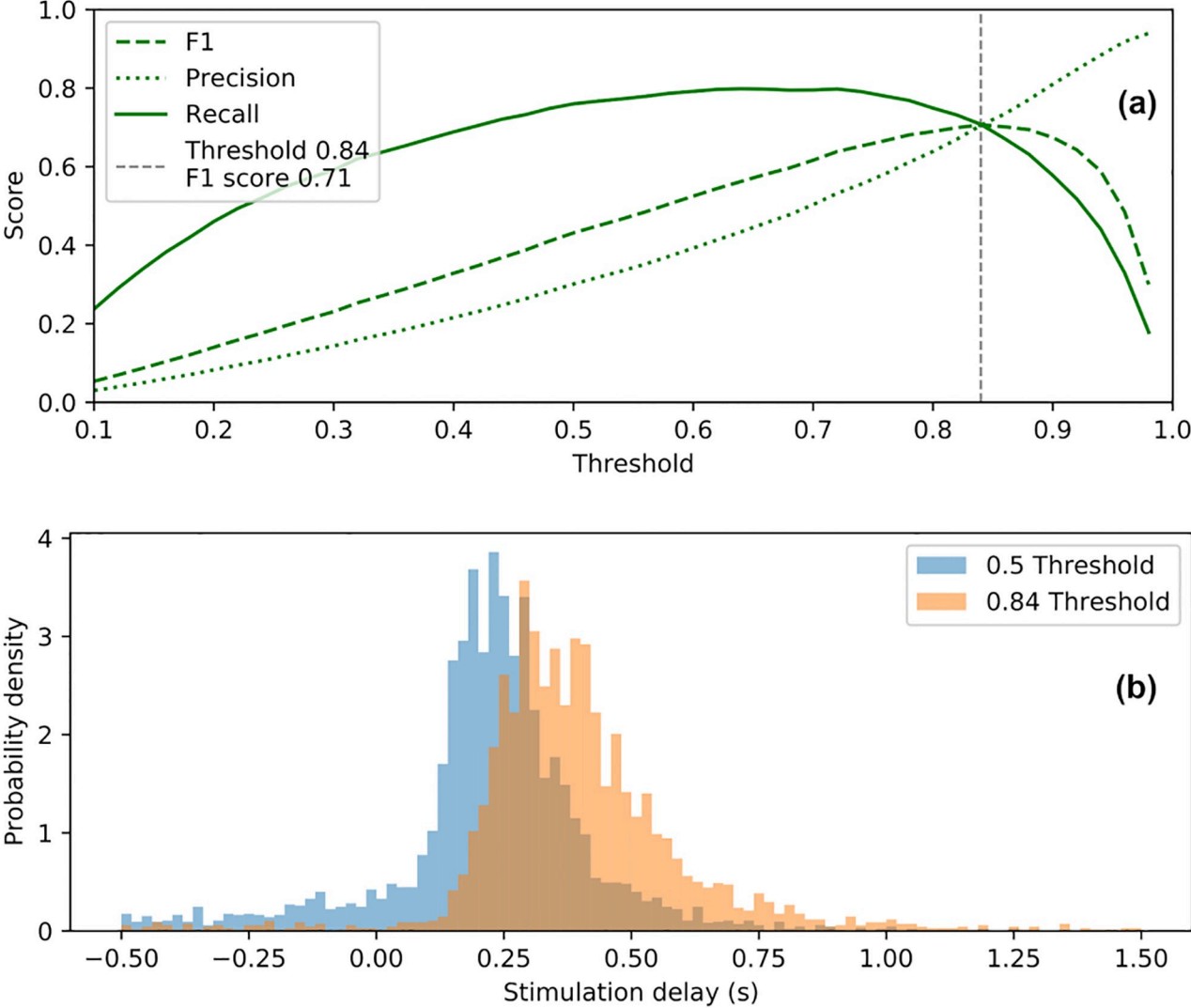

**Fig 6. Detection threshold trade-off.** (a) Evolution of the stimulation performance with respect to the chosen detection threshold on the ANN output value. A threshold of 0.84 yields the optimal trade-off; however, researchers may wish to select different parameters according to experimental objectives. (b) Distribution of stimulation delays for a classifier with 0.5 and 0.84 thresholds, respectively. Increasing the threshold yields longer delays. Note that delays are negative when spindles are stimulated in advance of human expert annotation.

relevance for the neuroscience community. Our resulting system is the first portable device to be able to detect and stimulate sleep spindles in real time with an f1-score of 0.71, measured on MODA, a dataset renowned for the reliability of its labels. The Portiloop system can be adapted to any application of EEG closed-loop stimulation, and potentially, any other neurophysiological signal. As opposed to classical heuristics, our deep learning-based approach does not require specific knowledge of the phenomenon of interest when defining the classifier, nor does it require a way to extract the relevant information. Instead, a large dataset of annotated signals suffices to derive a high-performance model that detects complex patterns such as sleep spindles.

Although we compare our architecture to a state-of-the-art sleep spindle detector (Spindle-Net), we did not have access to their weights and thus we could not compare their original

model with ours directly on the MODA dataset. Instead, we retrained their architecture from scratch on MODA, using our own pipeline. Contrary to the observations of Kulkarni *et al.* [29], we were able remove the envelope and power inputs without harming the performance of our models.

Concerning PMBO, the algorithm produces high-performance lightweight architectures, but we note that the predictions of the meta-learner are often near-constant in well-performing areas of the search space, suggesting that the meta-model could not further predict the software cost. In other words, we were unable to differentiate between the best-performing configurations of the neural network. We surmise that this is due to the large variance in model performance from one training session to another. This might be further improved by additional training. In future work, techniques such as Integrated Gradients [44] could be used to better understand the search space, and potentially fine-tune the ANN.

Explainable artificial intelligence techniques such as this may also help researchers to reveal unknown dependencies in neural activity, for example that a spindle might be preceded by another pattern of neural activity (see S1 File for an exploration of which parts of the signal are used by the neural network for classification).

In addition, while the MODA dataset provides high-quality labels, training on a larger dataset of similar quality would likely further improve the performance of our models. Expanding MODA is a relevant avenue for future work, as is implementing transfer learning techniques (i.e., tools that allow a trained network to adapt to a different environment), because the EEG acquisition and signal may differ somewhat from the training data or between individuals. Transfer can be achieved with techniques such as domain randomization [45]. Alternatively, a dataset can be collected on the Portiloop and annotated following the same protocol as MODA.

Long term, we intend to target specific portions of sleep spindles for stimulation (e.g., beginning, middle, end; or by oscillatory phase). This harder task will likely involve labeling these portions and developing more advanced RNNs/Transformers so as to consistently predict sleep spindles. Although our model does use information far back in time to make predictions, we believe that the main role currently played by the RNN is to accumulate information regarding whether the last few windows were spindles or not, rather than actually predicting the future (see S1 File). Such models will likely be more complex and computationally hungry, which is why the newer hardware implementation of the Portiloop integrates an embedded tensor processing unit (a powerful neural network accelerator). In general, finding an optimal model for a given Portiloop application involves either retraining our ANN, or re-executing PMBO to find a whole new architecture. Both activities can be done by interested practitioners using tools that accompany this work.

In sum, we hope that the Portiloop will help the neuroscience community explore brain functions, such as the role of sleep spindles in memory consolidation.

## Supporting information

**S1 File.**
(PDF)

## Acknowledgments

We thank Karine Lacourse for expert advice on spindle detection, and the MODA team for database access. Fig 1 uses icons from https://flaticon.com.

## Author Contributions

**Conceptualization:** Nicolas Valenchon, Yann Bouteiller, Hugo R. Jourde, Xavier L'Heureux, Milo Sobral, Emily B. J. Coffey, Giovanni Beltrame.

**Data curation:** Nicolas Valenchon, Yann Bouteiller.

**Formal analysis:** Nicolas Valenchon, Yann Bouteiller.

**Funding acquisition:** Emily B. J. Coffey, Giovanni Beltrame.

**Investigation:** Nicolas Valenchon, Yann Bouteiller.

**Methodology:** Nicolas Valenchon, Yann Bouteiller, Xavier L'Heureux, Milo Sobral.

**Project administration:** Emily B. J. Coffey, Giovanni Beltrame.

**Resources:** Emily B. J. Coffey, Giovanni Beltrame.

**Software:** Nicolas Valenchon, Yann Bouteiller, Milo Sobral.

**Supervision:** Emily B. J. Coffey, Giovanni Beltrame.

**Validation:** Nicolas Valenchon, Yann Bouteiller, Xavier L'Heureux, Milo Sobral, Emily B. J. Coffey, Giovanni Beltrame.

**Visualization:** Nicolas Valenchon, Yann Bouteiller.

**Writing – original draft:** Nicolas Valenchon, Yann Bouteiller, Hugo R. Jourde, Emily B. J. Coffey, Giovanni Beltrame.

**Writing – review & editing:** Nicolas Valenchon, Yann Bouteiller, Hugo R. Jourde, Emily B. J. Coffey, Giovanni Beltrame.

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
