## [Decision Letter · Decision Letter 0]

3 Jan 2022

PONE-D-21-32222The Portiloop: a deep learning-based open science tool for closed-loop brain stimulationPLOS ONE

Dear Dr. Bouteiller,

Thank you for submitting your manuscript to PLOS ONE. After careful consideration, we feel that it has merit but does not fully meet PLOS ONE’s publication criteria as it currently stands. Therefore, we invite you to submit a revised version of the manuscript that addresses the points raised during the review process.

Please, revise manuscript for clarity, you are requested to do substantial restructuring and shortening of the manuscript. Please, clearly describe  the goal and methods of the study, and provide the details of the design and data analysis.

For datasharing, please, use a scientific repository (e.g. Figshare, Harvard Dataverse,...)

More than half the figures (Figures 1, 3-7, 9) only highlight processes and methods. The rest of the figures cannot be interpreted without the main text. Therefore, the scope of analyses and presentation of the results required significant improvement.

Consider concatenating figures to reduce the number of figures.

The potential for a toolbox is evident, but such a toolbox’s utility and functionality are not demonstrated, but rather implied. The prototypical toolbox distracts from the analyses.

How could someone get one of these devices? Are readers expected to be able to build them, following your instruction (in which case the instruction would have to be much more detailed)?

It would be nice to see a picture of the device and how it can be used in the lab environment.

Reduced ANNs for maximizing Recall and Precision are not developed or analyzed. No reduced ANN matched SpindleNet’s Recall (% true positives). Reduced ANNs all had superior Precision (% positives found). Why? Three reduced models strike a balance between Recall and Precision with f1 scores of 0.61, but no model variations achieved the performance of expert raters. Why? All the reduced models perform essentially the same.

Minor comments: Table 1. Please spell out IExp, as this term / abbreviation is not used in the text. No stimulation is actually conducted in this study. Do the Authors mean actuate?

Do not use a special symbol (infinity) in the title, this will be very difficult for indexing, search engines etc.

We look forward to receiving your revised manuscript.

Kind regards,

Gennady S. Cymbalyuk, Ph.D.

Academic Editor

PLOS ONE

Journal Requirements:

Reviewers' comments:

Reviewer's Responses to Questions

**Comments to the Author**

1. Is the manuscript technically sound, and do the data support the conclusions?

Reviewer #1: Partly

Reviewer #2: No

2. Has the statistical analysis been performed appropriately and rigorously? 

Reviewer #1: I Don't Know

Reviewer #2: N/A

3. Have the authors made all data underlying the findings in their manuscript fully available?

Reviewer #1: Yes

Reviewer #2: Yes

4. Is the manuscript presented in an intelligible fashion and written in standard English?

Reviewer #1: No

Reviewer #2: Yes

5. Review Comments to the Author

Reviewer #1: This is a very long and non-stringent, yet potentially very interesting manuscript. I am not able to give an in-depth review at this point, since reading the manuscript is rather confusing, and it would probably take me several days to figure out everything that you are trying to say. I would advise you to do some substantial restructuring and shortening work, then I'll be happy to provide an in-depth review. For the moment, I have some mostly technical remarks:

* I would not recommend to use a special symbol (infinity) in the title, this will be very difficult for indexing, search engines etc.

* Also, it is not the gold standard to share data via a private Google Drive, have you considered using a scientific repository (e.g. Figshare, Harvard Dataverse,...)

* If my computer is counting correctly, the word count is ~10,000, which is clearly too long for the manuscript to be read by most people, including myself.

* This manuscript should not come with 13 figures. Since most of them contain very little information, consider concatenating them to three or four figures with several subplots.

* It is unclear to me what exactly you are presenting in this paper - you say it's about a new device, but at the same time you talk a lot about machine learning and software benchmarking fundamentals, and sleep physiology, which in my humble opinion is just too much for one manuscript. Please try to be concise and have a clear focus for the manuscript. For example, the best ANN architecture for spindle detection is in principle independent of any hardware implementation, and could therefore be dealt with in a separate paper (or in the Supplementary Material, if you do not deem it worth of a separate paper).

* Related to this: You open-sourced the software, but how could I get one of these devices if I wanted to? I may have missed it, but I found no information on this. Was it self-built by your lab or did you have an industry partner (which I don't assume since there is on conflict of interest declared). Can I buy it from you (in which case a COI would need to be declared), or am I expected to build it myself, following your instruction (in which case the instruction would have to be much more detailed)? Without at least mentioning any prospective way that I could get hold of one of these devices in the future, I'm not sure how useful a paper describing it will be.

* by the way, it would be nice to see a picture of the device and how it can be used in the lab environment.

* Related to this, I did not find any information about data storage for offline analysis, and how it would integrate with existing lab infrastructures (e.g. Labstreaminglayer).

To sum up, I think closed-loop stimulation is a relevant field with a lack of well-functioning and easy-to-use devices. I would therefore highly appreciate if you could bring your paper into a more digestable shape, so that your work can receive its due appreciation.

Reviewer #2: Summary:

The Authors developed hardware that can implement Artificial Neural Networks (ANNs) for detecting sleep spindles using FPGAs. Design constraints were imposed to make the hardware lightweight and portable for potential application in close-loop stimulation studies. This study focuses on a deep-learning, open-source toolkit for model-driven design of the FPGA ANNs, Portiloop, using a public sleep dataset (MODA) for training and testing. Such a toolkit could be useful for studies on sleep and brain stimulation, but the results are lacking. I have three concerns.

Concern 1 – The potential for a toolbox is evident, but such a toolbox’s utility and functionality are not demonstrated, but rather implied. The prototypical toolbox distracts from the analyses.

Concern 2 – Reduced ANNs for maximizing Recall and Precision are not developed or analyzed. No reduced ANN matched SpindleNet’s Recall (% true positives). Reduced ANNs all had superior Precision (% positives found). Why? Three reduced models strike a balance between Recall and Precision with f1 scores of 0.61, but no model variations achieved the performance of expert raters. Why? All the reduced models perform essentially the same.

Concern 3 – More than half the figures (Figures 1, 3-7, 9) only highlight processes and methods. The rest of the figures cannot be interpreted without the main text. Therefore, the scope of analyses and presentation of the results required significant improvement.

Minor comments:

Table 1. Please spell out IExp, as this term / abbreviation is not used in the text.

No stimulation is actually conducted in this study. Do the Authors mean actuate?

6. PLOS authors have the option to publish the peer review history of their article (what does this mean?). If published, this will include your full peer review and any attached files.

Reviewer #1: **Yes: **Marius Keute

Reviewer #2: No

---

## [Author Response · Author response to Decision Letter 0]

7 Mar 2022

Please see the attached PDF : Response to Reviewers.

transcript:

EDITOR:

---

Please, revise manuscript for clarity, you are re-

quested to do substantial restructuring and short-

ening of the manuscript.

>

Thank you for your suggestions. We have significantly

restructured the manuscript and have moved some de-

tails to Supplementary Information (please see response

to reviewers for specific changes). A challenge to our in-

terdisciplinary (neuroscience-engineering) work is that

the portion of collaborative projects that is not specific

to a reader’s discipline can be difficult for them to evalu-

ate and the details are perhaps less interesting to them.

However, the two parts are integral to one another: the

spindle detection application and its validation is not of

interest in fundamental neuroscience separately, but are

necessary for neuroscientists to support the device’s use

and validity in all subsequent works using it. Conversely,

the hardware and software designs are not conceptually

groundbreaking in isolation, but rather their adaptation

and combination in the service of fundamental science

is. We believe that both portions are necessary to in-

clude and in some level of detail, but have made the text

more readable, and have clarified how the work can be

useful to audiences with different foci early in the intro-

duction.

---

Please, clearly describe the goal and methods of

the study, and provide the details of the design

and data analysis.

>

We have described the goal and approach of the study

more clearly in the Introduction and in the Abstract

(e.g., from the Abstract: “In this article, we propose the

Portiloop, a deep learning-based, portable and low-cost

closed-loop stimulation system able to target specific

brain oscillations. We first document open-hardware

implementations that can be constructed from commer-

cially available components. We also provide a fast,

lightweight neural network and an exploration algorithm

that automatically optimizes the network model to the

desired brain oscillation. Finally, we validate the tech-

nology on a challenging test case of real-time sleep spin-

dle detection, with results comparable to off-line ex-

pert performance.Software and plans are available to

the community as an open science initiative to encour-

age further development and advance closed-loop neu-

roscience research”

REVIEWER 1:

---

I would not recommend to use a special symbol

(infinity) in the title, this will be very difficult for

indexing, search engines etc

>

The infinite sign has been replaced by normal charac-

ters.

---

Also, it is not the gold standard to share data via

a private Google Drive, have you considered us-

ing a scientific repository (e.g. Figshare, Harvard

Dataverse,...)

>

Our dataset, models, and all our relevant code are now

provided as part of the public Portiloop GitHub repos-

itory for convenience (this enables e.g. running our

trained neural network out-of-the box and visualizing

results in an interactive fashion). The private Google

Drive link has been removed

---

If my computer is counting correctly, the word

count is 10,000, which is clearly too long for the

manuscript to be read by most people, including

myself.

It is unclear to me what exactly you are present-

ing in this paper - you say it’s about a new device,

but at the same time you talk a lot about ma-

chine learning and software benchmarking funda-

mentals, and sleep physiology, which in my hum-

ble opinion is just too much for one manuscript.

Please try to be concise and have a clear focus for

the manuscript. For example, the best ANN ar-

chitecture for spindle detection is in principle in-

dependent of any hardware implementation, and

could therefore be dealt with in a separate paper

(or in the Supplementary Material, if you do not

deem it worth of a separate paper).

Related to this: You open-sourced the software,

but how could I get one of these devices if I

wanted to? I may have missed it, but I found

no information on this. Was it self-built by your

lab or did you have an industry partner (which I

don’t assume since there is on conflict of interest

declared). Can I buy it from you (in which case a

COI would need to be declared), or am I expected

to build it myself, following your instruction (in

which case the instruction would have to be much

more detailed)? Without at least mentioning any

prospective way that I could get hold of one of

these devices in the future, I’m not sure how use-

ful a paper describing it will be.

by the way, it would be nice to see a picture of

the device and how it can be used in the lab en-

vironment

>

We have considerably restructured and shortened the

manuscript, clarified the objectives, and have moved

many of the hardware details to Supplementary Infor-

mation and a Github repository. This is an interdisci-

plinary work, in the sense that its focus and novelty is

on the combination and adaptation of existing technolo-

gies to create a research tool for neuroscience. We have

provided guidance to readers to help them decide which

sections to which to direct their attention, depending

on their background and interests. For example, people

with the necessary engineering background to build a

device will be more interested in the hardware details

(note that details are moved to Github), a specialist in

neural networks wishing to use or adapt the network for

a different neural event might be most interested in the

neural network implementation, whereas neuroscientists

who are more interested in evaluating the device’s re-

search capabilities are likely to get more out of the case

study sections (please see the paragraph in the Introduc-

tion beginning with “The scope of this work is (...)”).

We have also clarified what is being offered and who

might be able to use it to what ends, throughout (e.g.,

hardware plans, code for the PMBO algorithm and for

the ANN), and we have added pictures of two Portiloop

hardware implementations, as suggested.

We hope our revised version is more clear about the

methodological focus of the manuscript, and that you

will enjoy giving it an in-depth review

---

Related to this, I did not find any information

about data storage for offline analysis, and how it

would integrate with existing lab infrastructures

(e.g. Labstreaminglayer)

>

Thank you for the suggestion; we have added the capa-

bility for streaming through LSL and have added in the

text, “The Portiloop comes with pre-installed software

for recording and visualizing the EEG signal, as well

as Python programming interface for the development

of extensions or new applications. Practically, the Por-

tiloop can be accessed via WiFi (it acts as a standalone

access point and it can connect to an existing network)

and it provides a web-based interface based on Python

Notebooks that allows to configure the EEG channels,

visualize the signals in real-time, start and stop EEG

recording (see Figure 3). The recording can be saved

either in the internal memory (32GB) or an SD card in

EDF format, or streamed through the network using the

Lab Streaming Layer (LSL), which guarantees synchro-

nization with microsecond accuracy.”

---

This manuscript should not come with 13 figures.

Since most of them contain very little informa-

tion, consider concatenating them to three or four

figures with several subplots.

>

This manuscript should not come with 13 figures.

Since most of them contain very little informa-

tion, consider concatenating them to three or four

figures with several subplots.

---

To sum up, I think closed-loop stimulation is a

relevant field with a lack of well-functioning and

easy-to-use devices. I would therefore highly ap-

preciate if you could bring your paper into a more

digestable shape, so that your work can receive its

due appreciation.

>

Thank your for your support. We hope you will find our

revised version much improved, and easy to digest

REVIEWER 2:

---

The potential for a toolbox is evident, but such a

toolbox’s utility and functionality are not demon-

strated, but rather implied. The prototypical

toolbox distracts from the analyses.

>

As noted also by Reviewer 1, the scope and goals of

the original manuscript were not clear. We have con-

siderably restructured and shortened the manuscript,

clarified the objectives, and have moved many of the

hardware details to Supplementary Information and a

Github repository. The current work focuses on meth-

ods development and proof-of-concept using a challeng-

ing test case. Our neural network training pipeline,

dataset and lightweight pre-trained models are now pub-

licly available on the Github repository of the project.

In particular, this enables comparison of our results and

using trained models out-of-the box, and we hope will

answer your concerns regarding utility and functional-

ity, at least for the software part.

The scope is now more clearly summarized in the Ab-

stract (please see also Introduction): “In this article, we

propose the Portiloop, a deep learning-based, portable

and low-cost closed-loop stimulation system able to tar-

get specific brain oscillations. We first document open-

hardware implementations that can be constructed from

commercially available components. We also provide a

fast, lightweight neural network and an exploration algo-

rithm that automatically optimizes the network model

to the desired brain oscillation. Finally, we validate the

technology on a challenging test case of real-time sleep

spindle detection, with results comparable to off-line ex-

pert performance. Software and plans are available to

the community as an open science initiative to encour-

age further development and advance closed-loop neu-

roscience research”

---

Reduced ANNs for maximizing Recall and Preci-

sion are not developed or analyzed. No reduced

ANN matched SpindleNet’s Recall (% true posi-

tives). Reduced ANNs all had superior Precision

(% positives found). Why? Three reduced mod-

els strike a balance between Recall and Precision

with f1 scores of 0.61, but no model variations

achieved the performance of expert raters. Why?

All the reduced models perform essentially the

same.

>

Please note that no model can achieve the performance

of expert raters for several reasons, most notably that

the expert ratings are what we use as our ground truth

and thus achieving their performance is statistically im-

possible. But furthermore, they have a fairly low inter-

rater agreement, which means that their ratings (that is

the closest we have to ground truth labels) are in fact

themselves pretty noisy. We have improved our expla-

nation of the challenges involved in labelling spindles

(see Section entitled “Offline sleep spindle detection for

labeling and performance comparison”).

We have also improved our explanation of the f1 score.

(“Once trained, the success of an algorithm on classify-

ing previously unseen data can be quantified using the

f1-score, which is a widely used metric to quantify an

average of recall (i.e., success in detecting events) and

precision (i.e., the proportion of detected events that

are correct), see Equation 1.”; a Figure 9 caption also

describes the trade-off with an illustration, “This pro-

vides a visualization of possible trade-offs in terms of

how many spindles we want to stimulate (recall) versus

how sure we want to be that all stimuli are relevant (pre-

cision). In terms of f1-score, the best such trade-off is

attained at a threshold of 0.84 with our model, yielding

a precision and a recall of 0.71.”)

Recall and Precision are meaningless metrics when con-

sidered independently from each other: if we predicted

that everything is a sleep spindle we would obtain a Re-

call of 100%, and if we predicted that only one data

point for which we are 100% sure to be in a sleep spin-

dle is indeed in a sleep spindle while everything else isn’t

we would obtain a Precision of 100%. The f1-score, on

the other hand, is a trade-off between these two metrics,

and it makes some sense to maximize it, contrary to the

Recall or Precision.

We do explore the optimal tradeoffs between Recall and

Precision in-depth in the submission, both in terms of

detection and real time stimulation (Figures 9, and in

Supplementary Information show different aspects of

this).

Concerning the reduced models, we conduct an ablation

study that shows the 1-input does indeed perform the

same as the 2-input (which is why we select the reduced

1-input model, see Table 1 rows (3) and (6)). Then,

we perform other ablations on this model only. The

reduced model trained without time-dilation performs

much worse (f1=0.47, see Table 1 row (7)) than the se-

lected model (f1=0.61, Table 1 row (6)). The subsequent

ablation is more subtle and shows that the “younger”

group (p1) contributes more to training than the “older”

group (p2) (see Table 1, rows (8-10)). We have clarified

these observations in the text

---

More than half the figures (Figures 1, 3-7, 9) only

highlight processes and methods. The rest of the

figures cannot be interpreted without the main

text. Therefore, the scope of analyses and pre-

sentation of the results required significant im-

provement.

>

Also at the suggestion of another reviewer, we have re-

duced the number of figures and moved some to Supple-

mentary Information. We have also adjusted the cap-

tions to be clearer as to the purpose of each figure, and

we believe they will be easier to interpret when placed in

the context of the text with their captions (as opposed to

the review format). We have considerably restructured

the manuscript to clarify the scope and aims (please see

Introduction, which has been largely re-written). The

figures do highlight processes and methods for the most

part, as these are the main advances in this interdis-

ciplinary work; the spindle application is intended as

a case study and validation, and to demonstrate the

behaviour of the system with respect to performance

trade-offs

---

Table 1. Please spell out IExp, as this term /

abbreviation is not used in the text

>

Thank you we have replaced it with Inter-rater agree-

men

---

No stimulation is actually conducted in this

study. Do the Authors mean actuate?

>

We agree that “actuate” would be a more general term.

However, given the proposed application in brain stim-

ulation and usage in that field, we prefer to keep the

stimulation terminology.

---

## [Decision Letter · Decision Letter 1]

27 Apr 2022

PONE-D-21-32222R1The Portiloop: a deep learning-based open science tool for closed-loop brain stimulationPLOS ONE

Dear Dr. Bouteiller,

Thank you for submitting your manuscript to PLOS ONE. After careful consideration, we feel that it has merit but does not fully meet PLOS ONE’s publication criteria as it currently stands. Therefore, we invite you to submit a revised version of the manuscript that addresses the points raised during the review process.

Please, provide actual examples of the EEG and how Portiloop classifies events, in this case, sleep spindles within the time series, and how this classification compares to expert manual classification. 

Abstract. Please, define and mention MODA in the Abstract.

Please, consider removing Figures 3.  In Figure 6, it will help to see the raw and processed signal, along with its envelope, and this could be integrated into Figure 2.

Table 2. The absence of visuals to support Portiloop's performance is a concern. The only figure that shows Portiloop’s processed EEG are in Figure 2. This paper needs a figure showing how Portiloop's different variations detect spindles in EEG compared to the nominal ground truth, the MODA dataset. MODA vs. 2-input Portiloop vs. 2-3 variants of Portiloop that favor speed over classification accuracy.

Table 2. Too many abbreviations and undefined terms in Row Set 2 -- on-line detection. Figures and tables should be interpretable without the main text. Therefore, the table needs a legend to describe, at a high level, what is meant by 1- and 2-inputs, ablation, td, p1, so forth.

Figure 7. As is, Figure 7 is abstracted from real run / compute times. It would helpful to contrast the classification performance before and after optimization. What is the trade-off between compute time and classification with a high software and hardware cost (without PMBO), versus a low software and hardware cost (with PMBO). Show EEG output of two ANNs off and on the Pareto front.

Figure 8. Figure 8 needs to be rotated clockwise 90 degrees.

Figure 9. It's not clear what score and threshold mean in this figure without referencing Figure 2 and the text. Figures should be mostly interpretable in isolation. Please, show a panel (as in Fig. 2) with the EEG colored and annotated to show what are true positives, false positive, etc. for stimulation timestamps. It may also help to see an example of a delay relative to the spindle (and its duration).

Please, consider whether you could bring the main text word count closer to 4000 or 5000 words and push more technical details to the Supplement or the Github repo. 

We look forward to receiving your revised manuscript.

Kind regards,

Gennady S. Cymbalyuk, Ph.D.

Academic Editor

PLOS ONE

Reviewers' comments:

Reviewer's Responses to Questions

**Comments to the Author**

1. If the authors have adequately addressed your comments raised in a previous round of review and you feel that this manuscript is now acceptable for publication, you may indicate that here to bypass the “Comments to the Author” section, enter your conflict of interest statement in the “Confidential to Editor” section, and submit your "Accept" recommendation.

Reviewer #1: (No Response)

Reviewer #2: (No Response)

2. Is the manuscript technically sound, and do the data support the conclusions?

Reviewer #1: Yes

Reviewer #2: Partly

3. Has the statistical analysis been performed appropriately and rigorously? 

Reviewer #1: I Don't Know

Reviewer #2: Yes

4. Have the authors made all data underlying the findings in their manuscript fully available?

Reviewer #1: Yes

Reviewer #2: Yes

5. Is the manuscript presented in an intelligible fashion and written in standard English?

Reviewer #1: No

Reviewer #2: Yes

6. Review Comments to the Author

Reviewer #1: The authors have done a good job with the revision. Objectives are laid out much clearer, and a number of well-made figures help the reader, and there is a comment about how to obtain such a device. There is also a My main concern with the manuscript, however, has not been adressed - it is still clearly too long to be read by a majority of the scientific community. While a part of the Methods section has been shifted to the Supplement, the word count for the main text is still between 9000 and 10000. I think the chances of having people actually read, and cite, the article would be much higher if the authors could bring the main text word count closer to 4000 or 5000 words and push more technical details to the Supplement or the Github repo. Also, the authors should be aware that the target audience are probably experimental neuroscientists more than computer scientists or electrical engineers, so a bit more information about the example study (sleep spindles) and a bit less technical detail would probably be appreciated.

Reviewer #2: This design paper covers Portiloop, a closed-loop and lightweight tool for event classification in timeseries using deep learning. The paper uses detection of sleep spindles from a public dataset, MODA, as a first test case and proof of concept. The manuscript, while dense in text, adequately describes the motivation and methods for constructing Portiloop. However, the results are left mostly in text form. Nontechnical readers will benefit from seeing actual examples of the EEG and how Portiloop classifies events, in this case, sleep spindles within the time series, and how this classification compares to expert manual classification. Therefore, my remaining concerns for this manuscript are in its presentation of the results. Some figures show flow chart of methods adequately described with text alone, and some figures need to show real examples of classified EEG data. My specific comments are below:

Abstract. I think it will help to define and mention MODA in the Abstract to give it a search link to a term relevant to sleep researchers.

Figures 3 and 6. I agree with Reviewer 1’s previous concern of too many figures. Figures 3 and 6 appear unnecessary. However, in Figure 6, it will help to see the raw and processed signal, along with its envelope, and this could be integrated into Figure 2.

Table 2. The absence of visuals to support Portiloop's performance is a concern. The only figure that shows Portiloop’s processed EEG are in Figure 2. This paper needs a figure showing how Portiloop's different variations detect spindles in EEG compared to the nominal ground truth, the MODA dataset. MODA vs. 2-input Portiloop vs. 2-3 variants of Portiloop that favor speed over classification accuracy.

Table 2. Too many abbreviations and undefined terms in Row Set 2 -- on-line detection. Figures and tables should be interpretable without the main text. Therefore, the table needs a legend to describe, at a high level, what is meant by 1- and 2-inputs, ablation, td, p1, so forth.

Figure 7. As is, Figure 7 is abstracted from real run / compute times. I think it would helpful to contrast the classification performance before and after optimization. What is the trade-off between compute time and classification with a high software and hardware cost (without PMBO), versus a low software and hardware cost (with PMBO). Show EEG output of two ANNs off and on the Pareto front.

Figure 8. Figure 8 needs to be rotated clockwise 90 degrees.

Figure 9. It's not clear what score and threshold mean in this figure without referencing Figure 2 and the text. Figures should be mostly interpretable in isolation. I recommend showing a panel (as in Fig. 2) with the EEG colored and annotated to show what are true positives, false positive, etc. for stimulation timestamps. It may also help to see an example of a delay relative to the spindle (and its duration).

7. PLOS authors have the option to publish the peer review history of their article (what does this mean?). If published, this will include your full peer review and any attached files.

Reviewer #1: **Yes: **Marius Keute

Reviewer #2: No

---

## [Author Response · Author response to Decision Letter 1]

11 Jun 2022

The submission has been revised substantially, please see the Response to Reviewers file (at the end of the generated PDF).

---

## [Decision Letter · Decision Letter 2]

16 Jun 2022

The Portiloop: a deep learning-based open science tool for closed-loop brain stimulation

PONE-D-21-32222R2

Dear Dr. Bouteiller,

We’re pleased to inform you that your manuscript has been judged scientifically suitable for publication and will be formally accepted for publication once it meets all outstanding technical requirements.

Kind regards,

Gennady S. Cymbalyuk, Ph.D.

Academic Editor

PLOS ONE

Additional Editor Comments (optional):

Reviewers' comments:

Reviewer's Responses to Questions

**Comments to the Author**

1. If the authors have adequately addressed your comments raised in a previous round of review and you feel that this manuscript is now acceptable for publication, you may indicate that here to bypass the “Comments to the Author” section, enter your conflict of interest statement in the “Confidential to Editor” section, and submit your "Accept" recommendation.

Reviewer #1: All comments have been addressed

Reviewer #2: All comments have been addressed

2. Is the manuscript technically sound, and do the data support the conclusions?

Reviewer #1: Yes

Reviewer #2: Yes

3. Has the statistical analysis been performed appropriately and rigorously? 

Reviewer #1: Yes

Reviewer #2: Yes

4. Have the authors made all data underlying the findings in their manuscript fully available?

Reviewer #1: Yes

Reviewer #2: Yes

5. Is the manuscript presented in an intelligible fashion and written in standard English?

Reviewer #1: Yes

Reviewer #2: Yes

6. Review Comments to the Author

Reviewer #1: I still think it is quite long, but otherwise I have no further complaints. I recommend to accept this paper.

Reviewer #2: While I feel it would have helped to show non-technical readers the explicit performance and tradeoffs between the different ANNs (e.g., examples of spindle detection on time series with maximal recall vs. maximal precision vs. maximal f1 vs. maximal hardware costs vs. optimal tradeoff in software and hardware costs etc.), this is stylistic and not essential. Thank you for addressing my technical concerns, and I have no further comments.

7. PLOS authors have the option to publish the peer review history of their article (what does this mean?). If published, this will include your full peer review and any attached files.

Reviewer #1: **Yes: **Marius Keute

Reviewer #2: No

---

## [Editor Report · Acceptance letter]

12 Aug 2022

PONE-D-21-32222R2 

The Portiloop: a deep learning-based open science tool for closed-loop brain stimulation 

Dear Dr. Bouteiller:

I'm pleased to inform you that your manuscript has been deemed suitable for publication in PLOS ONE. Congratulations! Your manuscript is now with our production department. 

Kind regards, 

on behalf of

Dr. Gennady S. Cymbalyuk 

Academic Editor

PLOS ONE